# Evaluating State-Level Prescription Drug Monitoring Program (PDMP) and Pill Mill Effects on Opioid Consumption in Pharmaceutical Supply Chain

**DOI:** 10.3390/healthcare11030437

**Published:** 2023-02-03

**Authors:** Amirreza Sahebi-Fakhrabad, Amir Hossein Sadeghi, Robert Handfield

**Affiliations:** 1Department of Industrial and Systems Engineering, North Carolina State University, Raleigh, NC 27606, USA; 2Department of Business Management, Poole College of Management, North Carolina State University, 2806-A Hillsborough St. Building 217, Raleigh, NC 27695, USA

**Keywords:** opioid crisis, PDMP, Pill Mill, difference-in-difference, policy analysis, pharmaceutical supply chain

## Abstract

The opioid crisis in the United States has had devastating effects on communities across the country, leading many states to pass legislation that limits the prescription of opioid medications in an effort to reduce the number of overdose deaths. This study evaluates the impact of two categories of PDMP and Pill Mill regulations on the supply of opioid prescriptions at the level of dispensers and distributors (excluding manufacturers) using ARCOS data. The study uses a difference-in-difference method with a two-way fixed design to analyze the data. The study finds that both of the regulations are associated with reductions in the volume of opioid distribution. However, the study reveals that these regulations may have unintended consequences, such as shifting the distribution of controlled substances to neighboring states. For example, in Tennessee, the implementation of Operational PDMP regulations reduces the in-state distribution of opioid drugs by 3.36% (95% CI, 2.37 to 4.3), while the out-of-state distribution to Georgia, which did not have effective PDMP regulations in place, increases by 16.93% (95% CI, 16.42 to 17.44). Our studies emphasize that policymakers should consider the potential for unintended distribution shifts of opioid drugs to neighboring states with laxer regulations as well as varying impacts on different dispenser types.

## 1. Introduction

The widespread abuse and addiction to opioids is a serious issue opioid crisis that has been covered widely in the popular press. Opioids are a class of drugs that include prescription painkillers and heroin [1]. The opioid crisis has become a major public health concern in the United States, and it has had devastating effects on communities across the country. It is estimated that opioids were a factor in over 75% of the 91,799 drug overdose deaths in 2020 [2], and the number of overdose deaths has been increasing in recent years. The crisis has been driven in part by the over-prescription of opioid painkillers [3], which has led to widespread misuse and addiction [4]. Efforts to address the crisis have included efforts to reduce the number of prescription opioids available (state-level opioid prescription limit) as well as increase access to treatment for addiction.

A state-level opioid prescription limit is a restriction on the amount of opioid medication that can be prescribed to a patient by a healthcare provider [5,6,7]. These limits are put in place in order to reduce the amount of opioid medication that is available for misuse and to help prevent the development of opioid addiction. The specific details of these limits vary from state to state [2], but they typically involve setting a maximum daily dose of opioid medication that can be prescribed as well as a maximum duration for which the medication can be prescribed. These limits are intended to help ensure that patients are only receiving the amount of medication that they need for pain management and to help prevent the over-prescription of opioids.

PDMPs are government-run systems that are used to track the prescribing and dispensing of certain medications [8], including controlled substances such as opioid painkillers. These programs are typically run at the state level, and they collect data from pharmacies and other dispensing organizations on the medications that are being prescribed and dispensed to patients [9,10]. The information collected by these programs is then made available to healthcare providers, who can use it to monitor patients’ medication use and to help identify potential cases of prescription drug abuse or misuse. The goal of these programs is to help reduce the over-prescription of opioid medications and to prevent the development of addiction. According to studies, PDMP has been linked to a 12% decrease in opioid-related deaths and a 10% reduction in opioid prescribing among patients with employer-sponsored insurance [11].

The opioid epidemic has been fueled in part by the actions of opioid manufacturers, who have been accused of misleading doctors and patients about the risks of these drugs and aggressively marketing them for uses that are not backed by scientific evidence [12,13]. As a result, many people have been prescribed opioids for chronic pain and other conditions, leading to widespread addiction and overdose deaths. In recent years, there has been a great deal of scrutiny on opioid manufacturers, and some have faced legal action for their role in the crisis. For instance, Ziedan et al. (2020) show that the implementation of a Pill Mill law decreased the sale of prescription opioids by around 33% [14]. However, there can be hardly found holistic studies evaluating the effect of state-level opioid regulations on the supplier side (distributors and manufacturers) as well as the consumers (patients and pharmacies).

Figure 1 shows the key elements of the pharmaceutical supply chain which refers to the series of steps and processes involved in the distribution and delivery of prescription and over-the-counter medications from manufacturers to patients [15]. It includes the production and distribution of pharmaceuticals by manufacturers, the distribution and sale of drugs by wholesalers and retailers, and the dispensing of drugs by pharmacies. The pharmaceutical supply chain is a complex system that involves multiple stakeholders, including manufacturers, wholesalers, pharmacies, and regulatory agencies [16,17].

We seek insights into the effectiveness of PDMP and Pill Mill legislation to restrict opioid addiction, using supplier data to explore the relative effectiveness of such policies on the opioid supply chain. Many state legislative policies are designed to reduce consumer consumption, targeting overdose rates, illegal consumption, and so on [18,19,20]. As a result, supplier-level data are not widely analyzed in understanding the effectiveness of these policies. One benefit of employing supplier data is the ability to determine the extent of policy impact on suppliers, which could be further used in designing policies for suppliers of opioids. Another advantage is that it may help policymakers determine the effectiveness of state-level policies on the flow of opioids from out-of-state suppliers and possibly develop the requirements for multi-state or national-level policies.

Research on the effectiveness of PDMP and Pill Mill laws has produced mixed results, with some studies finding that these laws are associated with reduced opioid prescribing and overdose deaths, while others have found no significant impact [21,22]. Some factors that may influence the effectiveness of these laws include the degree to which they are integrated into clinical practice, the extent to which they are used by prescribers and dispensers, and the level of support and training provided to those who use the system. These laws are typically designed to regulate prescribers, who are the last stage in the drug supply chain and are referred to as dispensers in this study. However, this research aims to evaluate the impact of these laws on the earlier stage of the supply chain, which is distributors. The study also aims to compare the two types of PDMP along with Pill Mill laws to determine which one is the most effective.

This study seeks to assess the efficacy of PDMP and Pill Mill laws on the Per Capita Pill Volume (PCPV) sold by two types of supply chain players: dispensers and distributors. The analysis includes 48 states in the United States, omitting Alaska and Hawaii. The study aims to evaluate the impact of these laws on different players and examine the effect of the laws on drug distribution flows within and between states that have implemented them. In this regard, a case study of the pharmaceutical supply chain of opioid drugs for the state of Tennessee (TN) and neighboring states including Kentucky (KY), Virginia (VA), West Virginia (WV), Ohio (OH), Indiana (IN), Illinois (IL), North Carolina (NC), South Carolina (SC), Georgia (GA), Alabama (AL), Mississippi (MS), Arkansas (AR) and Missouri (MO) is provided in Section 3. Additionally, the study uses a difference-in-difference method with a two-way fixed effect design to analyze the connection between in-state drug policies and out-of-state distributors, sales, and opioid use disorder.

## 2. Methodology

### 2.1. Overview

The Drug Enforcement Administration (DEA) maintains a database called the Automation of Reports and Consolidation Orders System (ARCOS) that records the authorized sales of controlled substances for both humans and animals in grams to medical facilities, pharmacies, and practitioners. Manufacturers and distributors are required to disclose information about their stocks, purchases, and dispositions of Schedule I and II drugs as well as Schedule III narcotic drugs and gamma-hydroxybutyric acid. In order to compare the potency of different prescription opioids, the annual sales of selected opioids (buprenorphine, codeine, fentanyl, hydrocodone, hydrophones, meperidine, methadone, morphine, and oxycodone) were converted into a common measurement called morphine equivalents [23,24]. The current version of ARCOS contains records from 2006 to 2014 and includes information about the amounts of opioids sold and the parties involved in the transactions. These data can be used to monitor and investigate the distribution of opioids and identify potential sources of abuse or diversion.

In addition to the ARCOS data, the study also uses demographic data as confounders from a previous study [6,25]. The ARCOS data are aggregated at the quarterly state level and combined with demographic data to enable more detailed analysis. The demographic data included the gender ratio, age groups, racial demographics, employment status, poverty level, population size, and whether or not Medicaid expansion had been adopted by the state in question. These data were obtained from population surveys. The study is conducted at the level of two opioid supply chain players: dispensers and distributors, while manufacturers are excluded. The reason for excluding manufacturers is that they are not diverse enough across states to be studied in difference-in-difference models. ARCOS data are aggregated to the quarterly state level as it gives the best precision for our analysis.

### 2.2. State Policies

In this study, we analyzed three categories of laws related to PDMPs with the aim of controlling the supply of opioid prescriptions. PDMPs are state-run databases that track controlled substances dispensed to patients, with the goal of identifying and preventing prescription drug abuse and diversion. These laws are implemented at four levels:1.PDMP access laws that provide access to any type of PDMP.2.Mandatory PDMPs that require prescribers to access the PDMP database before prescribing opioids under certain circumstances.3.Operational PDMPs, which are defined as having access to a modern system as the database.4.Electronic PDMPs, which means having access to an electronic database

We did not analyze the second category of laws in this study because the implementation dates for most of these laws are outside the time frame of the ARCOS data. Additionally, in many states, the implementation time of the first and last categories was often close to each other, resulting in the use of only ePDMP laws.

Another category of policies that is well known is called Pill Mills, in addition to PDMPs. Pill Mill laws are regulations designed to prevent the illegal prescribing and dispensing of prescription medications, particularly controlled substances such as opioid painkillers. These laws may include provisions related to prescribing practices, the distribution and sale of prescription medications, and the regulation of medical practices and pharmacies. Pill Mill laws vary by state and may be enforced by various agencies, such as state medical boards, law enforcement agencies, and DEA. The goal of these laws is to prevent the illegal and unethical prescribing and dispensing of prescription medications, which can contribute to the problem of prescription drug abuse and addiction.

We defined treatment indicators based on the dates of these policies. If a law was active in a given quarter, we assigned it a value of 1; otherwise, it was assigned a value of 0. These treatment indicators were used in our analysis to determine the effects of the laws on opioid supply.

### 2.3. Statistical Analysis

Researchers use the difference-in-difference method with a two-way fixed effect design to evaluate the effectiveness of public policies and interventions such as opioid laws. This method involves comparing the change in an outcome variable for a treatment group (which received the intervention or policy) to the change in the outcome variable for a control group (which did not receive the intervention or policy). The comparison is made both before and after the intervention was implemented, so that the difference in the changes can be attributed to the intervention [26,27,28,29]. In difference-in-differences (DID) analysis, the treatment effect is estimated using the following model:(1)$$Y=\beta_{0}+\beta_{1}T+\beta_{2}D+\beta_{3}TD+\epsilon$$
where *Y* is the outcome variable, *T* is the treatment status, *D* is a dummy variable indicating the time period (pre-treatment or post-treatment), and ϵ is the error term. The coefficient $\beta_{1}$ represents the treatment effect, and $\beta_{2}$ represents time effect, while the coefficient $\beta_{3}$ represents the change in the treatment effect over time.

This model can be extended to allow for individual-specific characteristics by adding a vector of individual-specific characteristics (*X*) to the model. It is equivalent to state demographics in our study:(2)$$Y=\beta_{0}+\beta_{1}T+\beta_{2}D+\beta_{3}TD+\beta_{4}X+\beta_{5}TX+\epsilon$$
where *X* is a vector of individual-specific characteristics, coefficient $\beta_{4}$ represents the effect of the individual-specific characteristics on the outcome in the absence of treatment, while coefficient $\beta_{5}$ represents the interaction effect between the treatment and the individual-specific characteristics.

In comparison, the model proposed by Sun and Abraham [30] allows for the treatment effect to vary over time and across individuals and requires a more generalized version of the assumption of parallel trends. The model is formulated as follows:(3)$$Y=\beta_{0}+\beta_{1}T+\beta_{2}X+\beta_{3}time+\beta_{4}TX+\beta_{5}timeX+\epsilon$$
where time is time. The coefficient $\beta_{1}$ represents the overall treatment effect, while the coefficients $\beta_{4}$ and $\beta_{5}$ represent the interaction effects between the treatment and the individual-specific characteristics and time, respectively. These interaction effects allow the treatment effect to vary over time and across individuals. In practice, the effects that change over time are analyzed by dividing them into specific time intervals during which the treatment effects take place, making it ideal for DiD designs that encompass several time intervals.

The main difference between the DID model and the model proposed by Sun and Abraham is that the DID model allows for a change in the treatment effect over time, but it does not allow for heterogeneity in the treatment effect across individuals. The model proposed by Sun and Abraham allows for both time-varying and individual-specific treatment effects. Model (Equation 3) is more plausible for the case of our study. This is because the policies being studied are implemented at different times in different states (shown in Figure 2), and each state implemented the policy separately from others in response to the opioid status in their state. In this case, the assumption of parallel trends, which is commonly used in DID analysis, may be problematic. This is because the outcomes of the treatment and control groups may have evolved differently in the absence of treatment due to the differences in the timing and implementation of the policies.

The proposed method is also used to test the following hypothesis:

**Hypothesis 1.** 
*State-level PDMP and Pill Mill laws are effective policies that reduce the overall consumption of opioids, as indicated by a decrease in trade between dispensers and distributors. Therefore, it can be concluded that these laws have the desired impact on supply chain players.*


**Hypothesis 2.** 
*State-level laws have varying levels of efficacy.*


**Hypothesis 3.** 
*State-level laws are primarily effective within the borders of the states that have implemented them, but they may unintentionally have negative effects on neighboring states.*


Three types of dependent variables and three types of policies are used in model (Equation 3): (1) PCPV: the amount of per capita pill volume bought by dispensers or sold by distributors; (2) In-State PCPV: the amount of PCPV transactions that are happening within the border of the state that implemented the law; (3) Out-of-State PCPV: the amount of PCPV transactions that are happening outside the border of the state that implemented the law. All analyses are conducted in R using package fixest, which is developed for fixed-effect estimation [31]. The significance level is set to 95%, and the overall policy effect sign is determined by the mean value.

## 3. Results

### 3.1. Overall

Previous research has shown that state-level legislation can result in patients with opioid use disorder seeking prescribers in states with less stringent regulations [32]. In the context of this study, this phenomenon is equivalent to obtaining drugs from out-of-state distributors or shifting drug sales to out-of-state dispensers. The state of Tennessee serves as an example of the potential impacts of PDMP laws on the distribution of controlled substances. Tennessee implemented electronic PDMP laws in 2003, operational PDMP laws in 2010, and mandatory PDMP laws in 2013. Figure 3 and Figure 4 show the trend of controlled substance distribution from Tennessee distributors to dispensers in neighboring states from 2006 to 2014. The top destination states for Tennessee distributors were Tennessee, Georgia, and Alabama. Before the implementation of operational PDMP laws in 2010, the trend of distribution to Tennessee dispensers was increasing. However, after the policy was implemented, the trend stabilized, suggesting the policy’s effectiveness. On the other hand, the distribution trend to Georgia and Alabama dispensers increased after the policy’s implementation, indicating that Tennessee distributors may have shifted their sales to these states. This phenomenon was also observed in other states, with the impact varying depending on the level of policies in place in those states. Figure 5 shows that for most states, there was no significant impact from the policy, but the flow of controlled substances to dispensers in Georgia, Alabama, and Arkansas increased while the flow to dispensers in Illinois and Indiana decreased. The different impacts may be related to the level of policies in those states. Overall, these findings suggest that state-level laws may have the unintended consequence of shifting the distribution of controlled substances to neighboring states, and policymakers should consider this potential drawback when implementing these laws.

Table 1 presents data on how different opioid policies (electronic PDMP, operational PDMP, and Pill Mill) affected the use of opioids by distributors and dispensers. It was found that electronic PDMP and Pill Mill policies had a negative impact, indicating that they were effective in reducing opioid use. In contrast, operational PDMP policies did not have a significant effect. Additionally, the impact of these laws was found to be greater on distributors than on dispensers, which is interesting because the policies are typically intended to provide dispensers (i.e., doctors, pharmacists, and other healthcare professionals) with information about their patients’ prescription drug use in order to improve the quality of care and to protect public health. The table also includes fixed effect covariates, which are variables that control for state-specific and time-dependent changes, in order to isolate the pure effects of the policies. The table supports the conclusion that both Pill Mill and electronic PDMP laws are effective in reducing opioid use at the supply level, and that the use of an electronic system may be particularly helpful in the context of PDMP. The table also includes statistical indicators such as the R-square value and the number of observations.

To further test our hypotheses, we will use other dependent variables in our DiD model to analyze the impact of different supply chain policies on the prescribing and dispensing of controlled prescription drugs both within and outside the state. The model proposed by Sun and Abraham allows us to analyze the dynamic effect of these policies over time, as shown in Figure 6 and Figure 7. The x-axis shows the quarters before and after the implementation of the policy, and the y-axis shows the average treatment effect (ATT) or policy effect. The figures are divided by policy type and the dependent variable, with red and blue dots indicating negative and positive effects, respectively, and the gray shaded area representing the 95% confidence level. The overall significance of the policy is calculated based on the weighted average of all post-policy ATTs, which is the same as the number reported by Table 1. In most cases, there is no clear trend in the data before the policy was implemented, supporting the validity of our analysis. Additionally, the confidence intervals tend to widen in later quarters, which is expected as it becomes increasingly difficult to accurately assess the policy’s impact over longer periods of time. Our analysis covers a time frame of two years before and five years after the policy’s implementation date, and the data are aggregated at the quarterly level for more accurate results and in line with previous research [6].

Dynamic effect analysis allows us to analyze the long-term effects of a policy on a particular issue. In this case, the policy being examined is a state-level law designed to control the dispersion of opioid drugs. The analysis in Figure 6 shows that the operational PDMP law, one of three laws being studied, has the least significant impact on the level of PCPV (a measure of opioid dispersion) and may even have an adverse effect in later quarters. In contrast, ePDMP shows a non-significant decreasing trend in PCPV, indicating that it is more effective at reducing opioid dispersion. Ultimately, Pill Mill laws had the greatest impact with a significant decrease in trend. Additionally, the analysis shows that in-state distributors are more affected by the policy, with the trends for the Pill Mill and ePDMP laws being significantly decreasing in the second row. (Note that the second row is the amount of opioid drugs coming from in-state distributors to the in-sate dispensers, while the third row is the flow from out-of-state distributors to in-state dispensers.) However, the laws have no impact on out-of-state distributors and may even increase the flow of drugs in some cases, highlighting a limitation of state-level laws in controlling activities outside of the state. Policymakers should carefully consider these findings when determining the most effective approach to reducing opioid distribution.

Following the same logic, the analysis of distributors in Figure 7 shows that the operational PDMP law is not very effective at reducing the flow of these drugs. However, when compared to the previous analysis of dispensers in Figure 6, it becomes clear that distributors are more affected by the policy. This is a positive outcome, but it is important to further investigate the reasons behind this effect. Our analysis also suggests that some distributors may have exhibited anticipatory behavior prior to the policy implementation. As a result, the overall estimated effect for this group may not be entirely reliable. It is worth noting that such behavior is not unexpected, as the policies were not specifically targeted toward distributors. However, it is important to note that the majority of the pre-policy impacts observed are not statistically significant, despite being negative. This suggests that anticipatory behavior may not have a significant impact on the overall estimated effect. The analysis also reveals that the flow of sales to both in-state and out-of-state dispensers is reduced after the implementation of a PDMP law. This indicates that in-state distributors reduce their sales, but the demand for in-state dispensers is not necessarily reduced. As a result, the mismatch between supply and demand is met by out-of-state distributors. Overall, these findings suggest that while PDMP and Pill Mill laws may be effective at reducing the flow of opioid drugs from in-state distributors, they may not be sufficient on their own to fully control the distribution of these drugs.

### 3.2. Dispenser Type Analysis

The ARCOS data set contains information about the distribution of controlled substances, such as opioid painkillers, by various types of dispensers, including retail pharmacies, chain pharmacies, and practitioners, which together account for over 80% of total transactions. PDMPs and Pill Mill laws are implemented to address the issue of prescription drug abuse and addiction by regulating the prescribing and dispensing of controlled substances. The effects of these laws on pharmacies and practitioners may vary depending on the specific provisions of the laws and how they are implemented. For example, pharmacies may face additional administrative tasks and restrictions on dispensing controlled substances, while practitioners may be required to review the PDMP before prescribing and follow specific guidelines for prescribing controlled substances. However, these laws may also help pharmacies and practitioners identify and prevent prescription drug diversion and make informed prescribing decisions, respectively. In this part of the study, we aim to apply a dynamic treatment framework to each of the retail pharmacies, chain pharmacies, and practitioners to evaluate the impact of these policies on each group.

The results of the study are presented in Figure 8, Figure 9 and Figure 10, which illustrate the different effects of laws on different types of dispensers. As previously mentioned, operational PDMPs had the least impact, while electronic PDMPs had no effect on the amount of opioids prescribed by practitioners, but they reduced the amount procured from in-state distributors and dispensed by pharmacies. This suggests that practitioners are not significantly affected by PDMP laws. In contrast, Pill Mill laws had a significant impact on practitioners, reducing the amount of opioids prescribed by them, including those obtained from out-of-state distributors. Thus, Pill Mill laws appear to be the most effective at the practitioner level. However, when considering the impact on pharmacies, the effect of Pill Mill laws varies. For retail pharmacies, it reduces the total dispensed volume and the amount procured from in-state distributors, but it has no effect on the volume of drugs obtained from out-of-state distributors. On the other hand, for chain pharmacies, the law only reduces the dispensed volume of drugs obtained from in-state distributors, but it has an adverse effect on the volume of drugs obtained from out-of-state distributors.

Overall, it can be concluded that policies have varying effects on pharmacies and practitioners. Pill Mill laws are most effective for practitioners, but policymakers should be cautious about implementing them for pharmacies, particularly chain pharmacies, due to their negative impact on the flow of out-of-state drugs into the state.

## 4. Discussion

Opioid overdose death rates in the United States have been increasing in recent years, leading many states to pass legislation that limits the prescription of opioid medications in an attempt to decrease the number of overdose deaths. The effectiveness of these laws in achieving this goal is debated, with some studies finding they are effective and others finding conflicting results [2]. These laws may also have unintended consequences, such as motivating patients to use alternative drugs that may be more dangerous or addictive. The study discussed here examines the impact of state-level laws on opioid consumption in pharmaceutical supply chains in the U.S., including the effects on out-of-state distributors and opioid use disorder. The authors use the ARCOS data set to analyze the impact of three types of laws related to PDMP: electronic PDMPs, operational PDMPs, and Pill Mills.

Results suggest that operational PDMPs are the least effective at reducing the flow of opioids, while Pill Mill laws and electronic PDMPs show a non-significant decreasing trend in PCPV (a measure of opioid distribution). The laws also had a greater impact on in-state distributors, with the trends for the Pill Mill and ePDMP laws being significantly decreasing. However, the laws had no impact on out-of-state distributors and may even increase the flow of drugs in some cases, highlighting a limitation of state-level laws in controlling activities outside of the state. The authors also found that the PDMP and Pill Mill laws were effective at the level of distributors, leading to a reduction in their sales to in-state and out-of-state dispensers. Finally, the authors found that PDMP laws had different impacts on various dispenser types, with operational PDMPs having the least impact and electronic PDMPs having no effect on reducing the amount prescribed by practitioners but reducing the amount of opioids procured from in-state distributors and dispensed by pharmacies. Pill Mill laws significantly reduced the amount of opioids prescribed by practitioners and provided by out-of-state distributors, and they had varying effects on the dispersion of drugs by pharmacies and chain pharmacies.

In summary, the authors suggest that policymakers should consider the following when implementing state-level opioid policies, particularly PDMP and Pill Mill laws:1.Different policies may be more effective for different types of dispensers, and there is no one-size-fits-all solution. For example, based on the findings of the study, Pill Mill laws may be more effective in reducing Per Capita Pill Volume (PCPV) sold by practitioners, while Electronic PDMP (ePDMP) and Operational PDMP (OP-PDMP) laws may be more effective in reducing PCPV sold by pharmacies. As our results suggest, it is important for policymakers to consider the unique characteristics of different types of dispensers and tailor policies accordingly to achieve the desired outcome. These findings are in line with previous studies that show pharmacists and primary care physicians value the information provided by prescription drug monitoring programs, and that efforts to improve these programs should focus on making it easy for them to access all relevant information about a patient’s controlled substance prescriptions at the time they are making treatment decisions [33].2.The study has highlighted the potential for unintended consequences of opioid-related policies. For instance, it suggests that as a result of these laws, there may be an increase in opioid sales or procurement from out-of-state dispensers or distributors. This could lead to an increase in opioid use disorder in neighboring states. Therefore, policymakers should be cognizant of these potential spillover effects and take steps to mitigate them. Previous research has also shown that individuals who frequently obtain opioid prescriptions tend to travel farther and go to different states to fill them. Additionally, they disproportionately contribute to the overall number of opioid prescriptions dispensed. These findings underline the need for sharing information among programs that track prescription data in order to curb excessive opioid procurement [34].3.Policies may have impacts beyond their intended targets and may be more effective for different groups than originally intended. Policymakers should consider whether these unintended impacts are positive or negative and how they align with their goals. For example, while a law aimed at reducing opioid prescribing may be successful in achieving that goal, it may also have negative impacts on patients who depend on opioids for pain management. Policymakers should take a holistic approach when evaluating the impact of these laws and consider all potential consequences. Additionally, policymakers should continuously monitor and evaluate the laws’ effectiveness and make necessary adjustments to achieve the desired outcome.

## 5. Conclusions

In conclusion, the effectiveness of state-level laws aimed at reducing opioid prescribing and overdose deaths in the United States is mixed and varies depending on the specific details of the legislation and the type of dispenser being affected. The authors of this study found that operational PDMPs were the least effective at reducing the flow of opioids, while Pill Mill laws and electronic PDMPs showed a non-significant decreasing trend in prescription controlled substance volume. The laws also had a greater impact on in-state distributors, with the trends for the Pill Mill and ePDMP laws being significantly decreasing. However, the laws had no impact on out-of-state distributors and may even increase the flow of drugs in some cases, highlighting a limitation of state-level laws in controlling activities outside of the state. Policymakers should carefully consider the potential unintended consequences and impacts beyond the intended targets of each policy when implementing state-level opioid legislation. It should be noted that the results of this study are specific to the United States context and may not be directly generalizable to other countries or regions, but the model and methodology used in this study could potentially be adapted to address similar issues in other countries.

Our research suggests that studying the supply chain of drugs can provide insight into the effectiveness of state-level regulations and help decision-makers anticipate the potential impact of such regulations. By understanding the supply chain, it is possible to identify the sources of drugs and the ways in which they are being distributed, which can inform policy decisions and help prevent negative outcomes such as drug overdoses. This approach may be particularly useful for policymakers who are considering enacting new laws or regulations in an effort to address the opioid crisis or other public health issues related to drug abuse.

## Figures and Tables

**Figure 1 healthcare-11-00437-f001:**
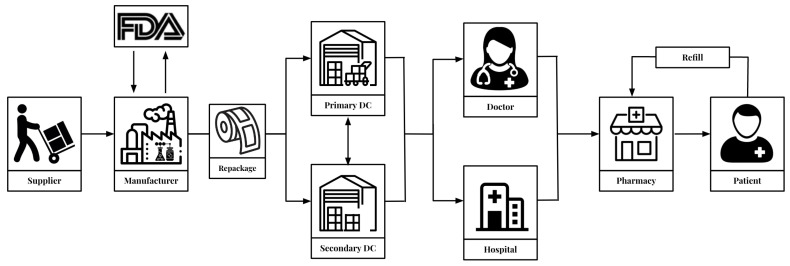
Pharmaceutical supply chain.

**Figure 2 healthcare-11-00437-f002:**
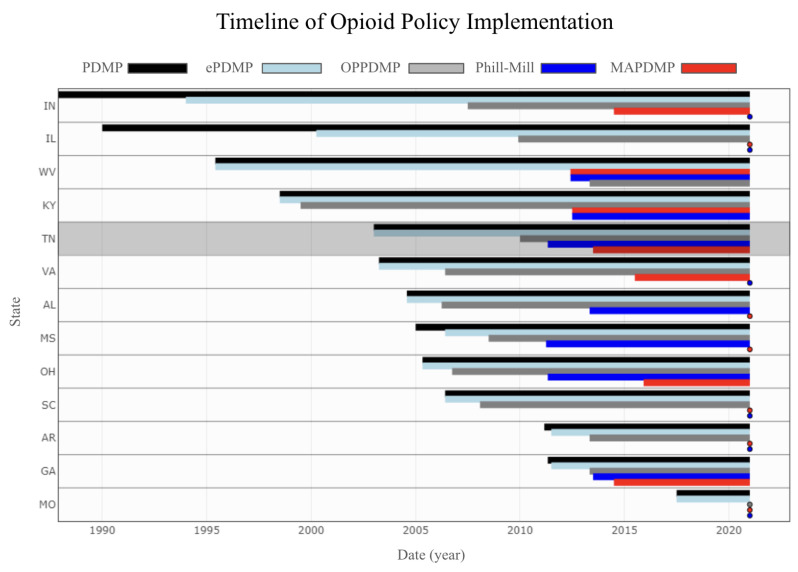
Policy Timeline; data on the policy implementation is provided by the RAND Corporation.

**Figure 3 healthcare-11-00437-f003:**
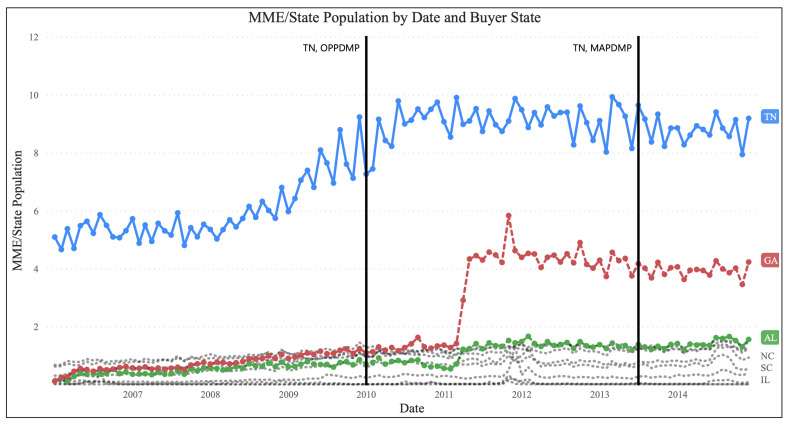
Trend of PCPV, excluding manufacturing buyers; source: ARCOS.

**Figure 4 healthcare-11-00437-f004:**
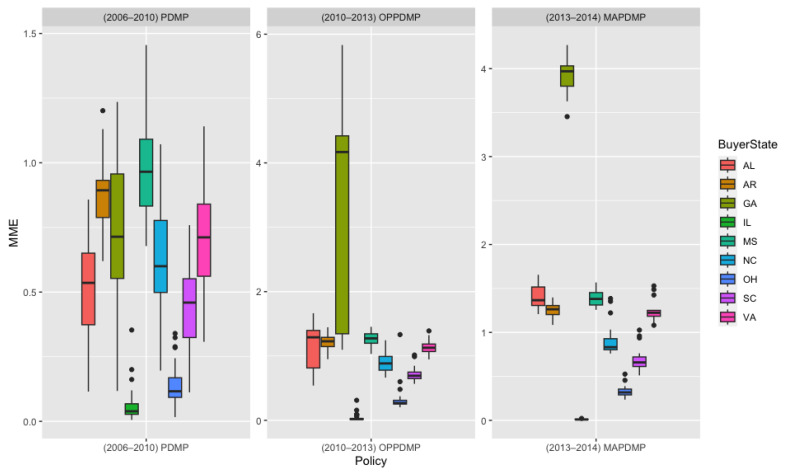
Amount of PCPV from distributors in TN to neighbor states after implementation of each PDMP policy in TN, excluding manufacturing buyers Source: ARCOS.

**Figure 5 healthcare-11-00437-f005:**
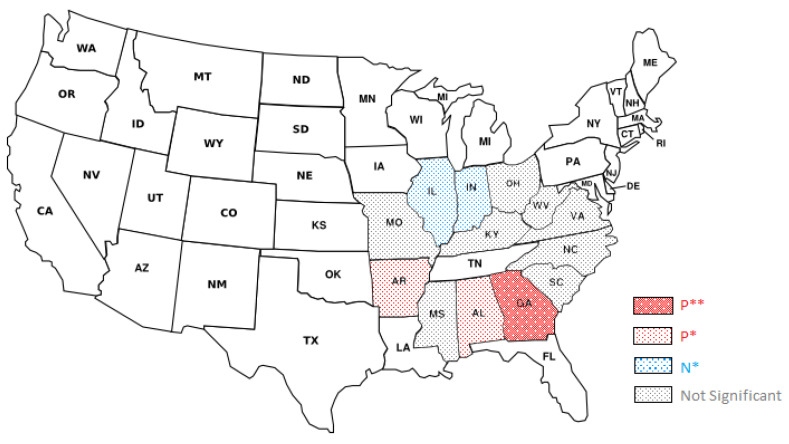
Amount of opioid drugs shipped from distributors in TN to neighbor states per destination capita after OP-PDMP implementation in TN, excluding manufacturing buyers; Signif. Codes: **: 0.05, *: 0.1, P: Positive Correlation, N: Negative Correlation; Source: ARCOS.

**Figure 6 healthcare-11-00437-f006:**
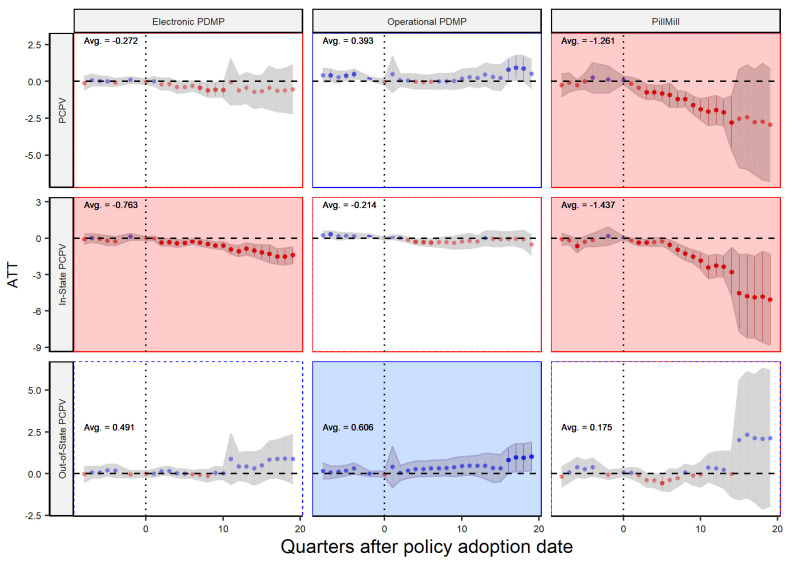
Policy Effect—Player: Dispenser.

**Figure 7 healthcare-11-00437-f007:**
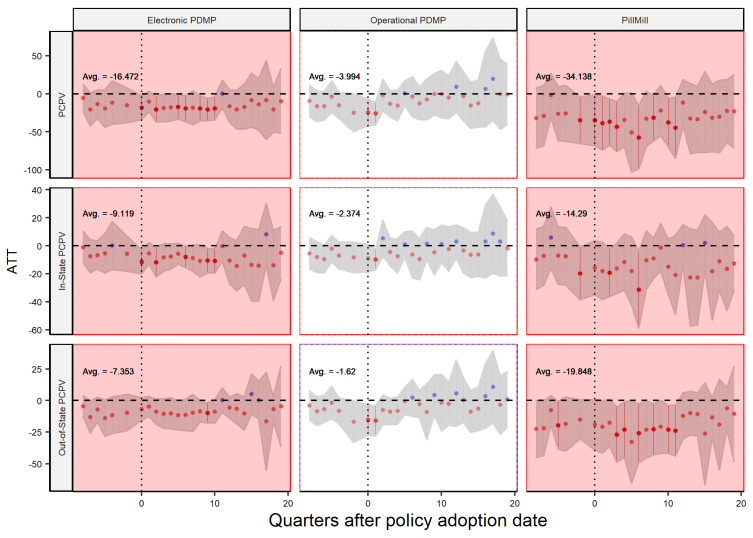
Policy Effect—Player: Distributor.

**Figure 8 healthcare-11-00437-f008:**
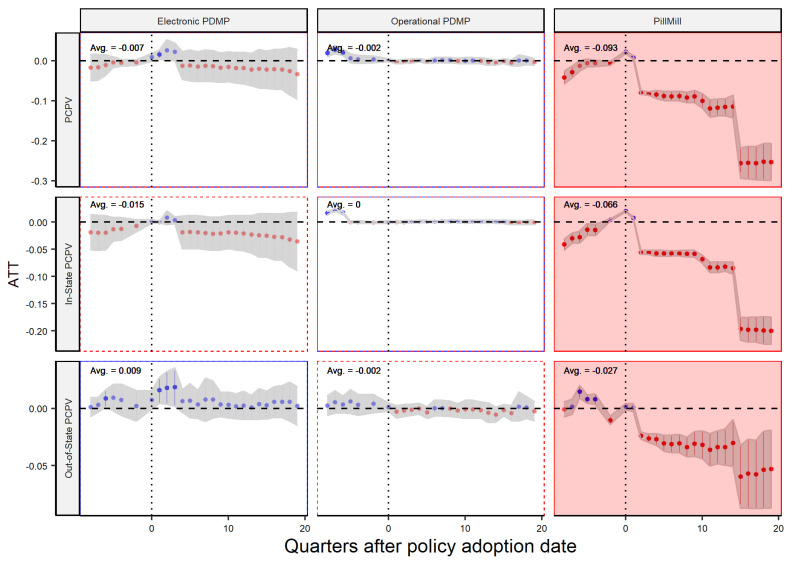
Policy Effect—Player: Practitioner.

**Figure 9 healthcare-11-00437-f009:**
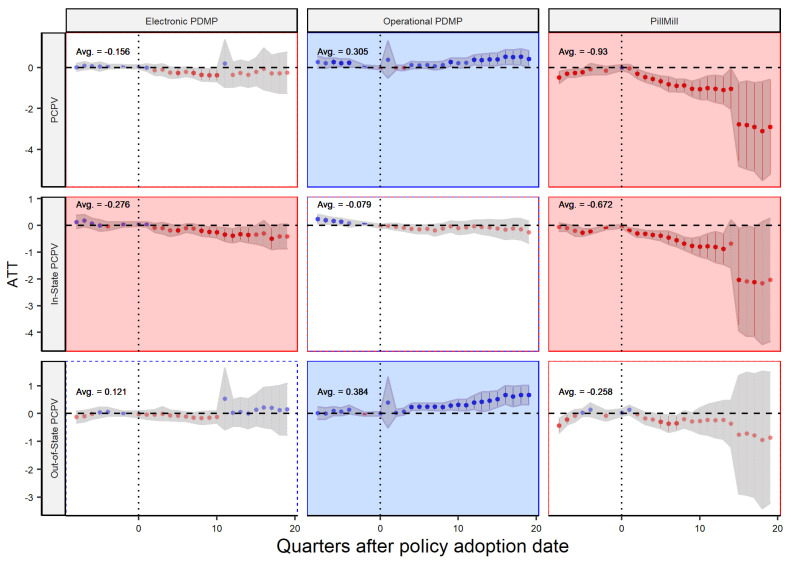
Policy Effect—Player: Retail Pharmacy.

**Figure 10 healthcare-11-00437-f010:**
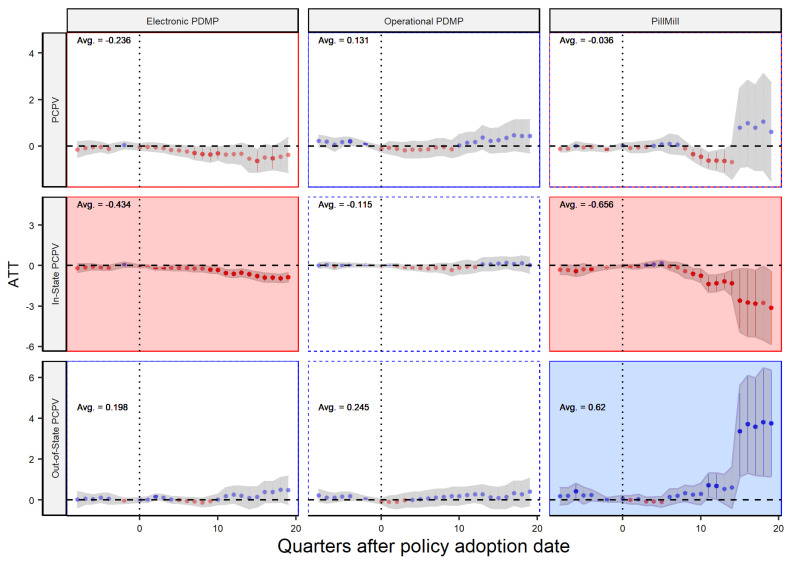
Policy Effect—Player: Chain Pharmacy.

**Table 1 healthcare-11-00437-t001:** PDMP and Pill Mill policies impact on opioid distributors and dispensers as the main supply chain players. The impact is evaluated through DiD method over PCPV bought (sold) by the dispenser (distributor).

	*Supply Chain Player* *Policy*					
	* **Dispenser** *			* **Distributor** *		
	* **Pill Mill** *	* **ePDMP** *	* **opPDMP** *	* **Pill Mill** *	* **ePDMP** *	* **opPDMP** *
* **Variables** *						
*ATT*	−1.3 *** (0.44)	−0.27 (0.49)	0.39 (0.28)	−34.1 *** (15.1)	−16.5 ** (6.6)	−4.0 (10.4)
* **Fixed-effects** *						
*State*	yes	yes	yes	yes	yes	yes
*Quarter*	yes	yes	yes	yes	yes	yes
* **Fit statistics** *						
$R^{2}$	0.956	0.951	0.975	0.918	0.918	0.918
*Observations*	1368	1475	1474	1401	1402	1402

Clustered (State) standard-errors in parentheses; Signif. Codes: ***: 0.01, **: 0.05.

## Data Availability

Data supporting reported results can be found, including links to publicly archived datasets analyzed or generated during the study at https://www.slcg.com/opioid-data/ (accessed on 15 August 2022).

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
