# Peer review of "Evaluating State-Level Prescription Drug Monitoring Program (PDMP) and Pill Mill Effects on Opioid Consumption in Pharmaceutical Supply Chain"

_healthcare, 2023, doi:10.3390/healthcare11030437_

Round 1

Reviewer 1 Report

This is a good research paper on a timely and important topic.

My main question about content is: 

--In Figure 7, it seems the negative red dots precede the policy date. Do the authors see this as anticipatory, and if so, would that reduce the estimated ES?

A few minor issues with wording: 

--In addition to the ARCOS data, the study also uses demographic data /to control for confounders/ identified in a previous study [6,23? ].

--Researchers used the difference-in-difference method

--The analysis includes 48 states in the United States, omitting Alaska and Hawaii

--seeking out prescribers in states with less stringent regulations[]. (cite?)

--Three types of dependant dependent variables and three types of policies

--happening outside the boarder border of the state

--greater on distributors than on dispensers which/,/ is interesting

--statistical indicators such as the R-squared value

--Table 1. PDMP and Pill Mill policies impact on opioid distributors and disperners dispensers as the main supply chain players. The impact is evaluated through DiD mothed method over PCPV bought (sold) by the dispenser (distributor)

--To further test and confirm our hypotheses

--all post-policy ATTs which/,/ is

--and the data is aggregated at the quarter/ly/ level (or better yet, aggregated by quarter)

--Sometimes, the phrase "Pill MIll laws" is capitalized, and other times, not. 

--Sometimes, "policy makers" is one word, other times, two words. 

--In the Discussion section, "The authors find that..." is a phrase that might be substituted with "we found," or "results suggest..."

Author Response

Dear Reviewer,

Thank you very much for your strong comments and valuable suggestions on our manuscript. These were very helpful for revising and improving our paper. We carefully studied your comments and made all the required changes. Please, find the point-by-point response to your comments attached. We hope for your approval of the revised version of our paper.

Reviewer 2 Report

The publication raises an interesting area. There are still few publications in this topic. The paper requires small editorial corrections.
1. remove the "?" at reference position [6,23] - page 3
2. Figura 2 and 5, add extra line spacing

Moreover section "3.1 Overall" indicates that the study covers 50 states. In my opinion, it is important to specify (more precise explanation) which analyzes cover all states, and which analyzes concern the TN state and neighboring states.
In my opinion, the manuscript is appropriate for the special issue indicated: Health and Medical Policy in the Era of Big Data Analytics.

Author Response

Dear Reviewer, 
Thank you very much for your strong comments and valuable suggestions on our manuscript. These were very helpful for revising and improving our paper. We carefully studied your comments and made all the required changes. Please, find the point-by-point response to your comments in the attached file.

We hope for your approval of the revised version of our paper.

Reviewer 3 Report

Please find my comments in the attached document.

Author Response

(The authors gave the same response as above.)

Round 2

Reviewer 3 Report

I would like to thank the authors for carefully addressing all of my concerns. I feel that the quality of the manuscript have been substantially improved and that that the article is - in its present form - suitable for publication in Healthcare.